# Evidence for a composite volcano on the rim of Jezero crater on Mars
Sara C. Cuevas-Quiñones[1,2], James J. Wray [1] ✉, Frances Rivera-Hernández[1] & Jacob B. Adler [1,3]

The Perseverance rover is currently exploring Jezero crater to collect, characterize and cache the first planned samples of Mars for future return to Earth. Orbital and rover observations suggest a volcanic origin for crater floor materials, sources of which have thus far been unknown. Here we describe a mountain on the crater's southeastern rim with morphological, thermophysical, and mineralogical properties consistent with explosive volcanoes previously identified on Mars and Earth. The mountain's low thermal inertia and scarcity of superposed impact craters are consistent with a fine-grained, weakly consolidated material such as volcanic ash. Possible flow margins from its northwestern flank extending onto Jezero crater's floor indicate that it could have plausibly supplied volcanic materials to the crater. If so, then radioisotope dating of igneous rock samples cached by Perseverance could eventually make this the first volcano of precisely known age on another terrestrial planet.

NASA's Perseverance rover landed in Jezero crater on Mars (Fig. 1a, b) in February 2021. The rover is exploring the geology, past habitability, and potential for life in this crater[1]; it is also collecting the first martian samples planned for return to Earth, a campaign broadly agreed to be the highest scientific priority for NASA's planetary robotic program in this decade[2]. Jezero crater was targeted[3] mainly due to its fluvio-lacustrine history, with an outlet valley indicative of fluid filling and overflow, and fan-delta deposits preserved at the mouths of two inlet valleys[4–8]. These deltaic sediments were viewed as ideal for concentration and preservation of ancient organic matter[9], and orbital detections of both clay and carbonate minerals within the crater confirmed that the products of water-rock interactions would be readily accessible to a surface mission[10–15]. Craters superposed on the watershed feeding these deposits suggest that aqueous activity took place ~3.8 Ga[16], with the final flows that formed the deltas ending by ~3.5 Ga[17].

A different set of geologic materials span most of Jezero crater's floor. Variably outcropping around the crater floor margins is a light-toned rock unit exhibiting spectral signatures of olivine, Mg-carbonate, and hydration; pre-landing hypotheses for its origin included lacustrine sedimentary fill[13], aeolian sandstone[14], or volcanic ash[18]. Rover observations have instead revealed this unit (Séítah formation) to be a lightly altered olivine cumulate rock, formed in an igneous intrusion or thick lava flow[19–21]. Whereas this light-toned rock unit predates the deltas[13,18,22], a darker-toned rock unit of more uncertain age dominates the central crater floor, with spectral signatures of pyroxene and a thickness up to ~10 m at its margins[13]. This darker-toned unit was originally interpreted as volcanic[5,13,23], but fluvial or

aeolian sedimentary origins were also suggested by the Perseverance science team prior to landing[18,24]. Perseverance observations have since found this unit (Máaz formation) to be igneous as well, a combination of lava flows and likely some pyroclastic layers[25,26].

Perseverance collected multiple samples of Jezero's Séítah and Máaz formations[27]; returning a subset of these to the Earth will allow precision dating of these igneous rocks, previously possible only for martian meteorites of unknown source locations[28]. Returned samples will have better geologic context, but the precise source of the volcanic rocks on Jezero's floor remains an open question. Most pre-landing studies of the crater floor either stated that no volcanic structure or vent that could represent such a source is apparent anywhere within or near Jezero[13,18] or did not address the issue. Yet on the crater's southeastern rim there is a conical edifice with a summit crater rising nearly 2 km above the surrounding plains. This mountain, recently named Jezero Mons, dominates the southeastern horizon in Perseverance rover images[29] (Fig. 1c). Horgan et al.[14] suggested it could be volcanic and thus a candidate source of volcanic rocks within Jezero crater. Here we evaluate this hypothesis using orbital datasets and comparisons to other volcanoes on Earth and Mars.

## Results

Jezero Mons (centered at 78.2°E, 18.2°N) was previously mapped as part of a "dusty, massive basement" unit[13], likely in part due to its lower thermal inertia compared to other major landforms in or around Jezero crater (Fig. 1d). We measured an average thermal inertia of 239 J m$^{-2}$ K$^{-1}$ s$^{-½}$ across

[1]School of Earth and Atmospheric Sciences, Georgia Institute of Technology, Atlanta, GA, 30332, USA. [2]Department of Earth, Environmental & Planetary Sciences, Brown University, Providence, RI, 02912, USA. [3]School of Earth and Space Exploration, Arizona State University, Tempe, AZ, 85287, USA.
✉e-mail: jwray@gatech.edu

**Fig. 1 | Overviews of Jezero Mons.** The mountain is ~21 km across for scale. **a** Topography of the eastern hemisphere of Mars, showing Jezero and other landmarks mentioned in the text. **b** Overhead view (CTX mosaic) of Jezero crater ($D \sim 45$ km). **c** Southeastern horizon viewed from NASA's Perseverance rover in western Jezero crater (Mastcam-Z "Van Zyl Overlook" Panorama, acquired on sols 53–63). **d** Oblique view from north-northeast of Jezero crater, showing fine-grained materials in blue and purple colors vs. coarser and/or indurated materials on the surrounding ancient highlands in yellow and orange (THEMIS nighttime IR mosaic colorizing THEMIS daytime IR brightness, with topography exaggerated ~3x; temperature scale based on THEMIS image I01664003). **e** Oblique view from southwest, showing the irregular summit crater and possible flows into Jezero crater (CTX visible imagery draped over MOLA topography exaggerated ~3x, colorized by elevation). Inset is a topographic profile (exaggerated ~3x) of the approximate cross-section visible in panel d (transect starting ~68° east of north from summit crater's center).

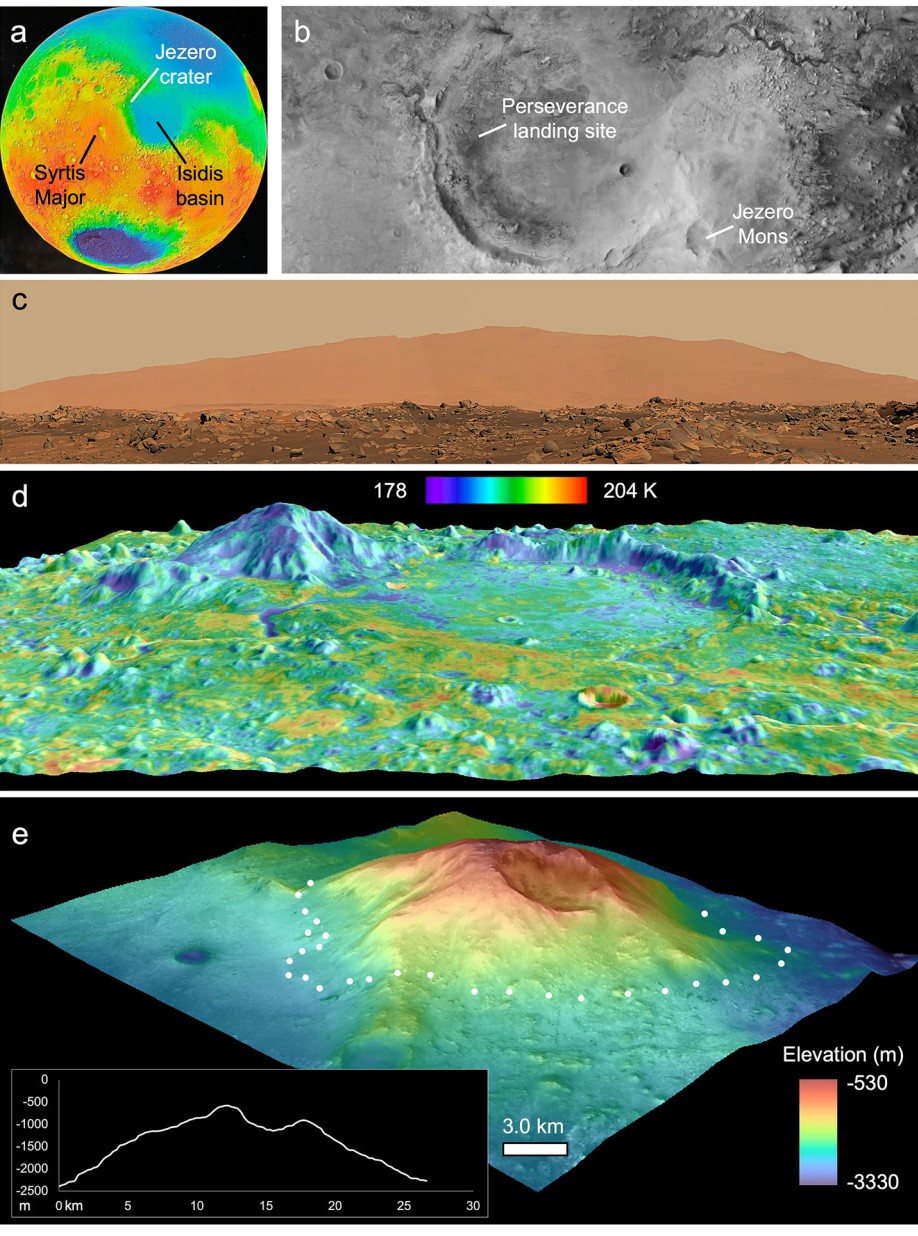

Jezero Mons, consistent with fine sand[30] or alternatively with even finer-grained material that is partially indurated or heterogeneously distributed across a surface that also includes rocky material[31]—indeed, values range from 136 to 444 J m$^{-2}$ K$^{-1}$ s$^{-½}$. Higher inertias typically occur along topographic ridges, including around the rim of an irregular crater that extends from the mountain's summit to approximately halfway down its southern flank (Fig. 1e). The polygonal outline of this crater's rim has segments (and some ridges inside it) mostly oriented in one of two orthogonal directions (Fig. 2a, c). The summit crater spans in width ~7 km from north to south and ~5 km from east to west, while the mountain overall is ~21 km across, nearly half of Jezero crater's diameter.

Infrared hyperspectral mapping (see Methods) of the northern and eastern flanks of Jezero Mons shows widespread pyroxene-bearing materials across the edifice (Fig. 2a). Exposures nearest the summit show a very broad spectral absorption centered near 2.0 μm (Fig. 3, spectra 1–2), consistent with a mixture of low- and high-calcium pyroxenes[32]. The strongest signatures are found where dark-toned outcrop is exposed that has a bluish appearance in enhanced-color images (Fig. 4a, b).

Alteration minerals are also observed in scattered locations. One kilometer-scale exposure on the northeast flank has spectral absorptions at

~1.4, 1.9, 2.3, 2.4, 2.5, and 3.9 μm (Fig. 3, spectra 6–7), consistent with a mixture of Fe/Mg-smectite and Mg-carbonate (and possibly olivine, based on spectral curvature from 1 to 2 μm), as widely observed in Jezero and the surrounding region[10–14]. These materials appear localized to light-toned outcrops oriented radially around a quasi-circular filled depression (Fig. 4c) and thus may be impact ejecta. One knob near this location shows a distinct set of absorptions at 1.39, 1.90, and 2.2 μm (Fig. 3, spectrum 3), consistent with hydrated opaline silica[33].

Small hydrated mineral exposures in and adjacent to the summit crater instead have broad absorptions at ~1.9 and 2.32 μm (Fig. 3, spectra 4–5), consistent with Fe/Mg-phyllosilicates. Relative to the flank smectite-carbonate spectra, these summit spectra lack 2.4 μm absorptions, have more asymmetric ~2.3 μm absorptions centered at longer wavelengths, and one example (spectrum #4) has an additional 2.25 μm absorption, all of which are consistent with partial chloritization of smectites[34]. These materials are commonly associated with light-toned outcrops, in some cases occupying topographic lows between pyroxene-rich ridges (Fig. 4a).

The contact between the pyroxene-rich Jezero Mons flank materials and underlying bedrock is well exposed in several areas (Fig. 5). Smooth,

**Fig. 2 | Detailed view of Jezero Mons.** Arrows point to locations of spectra (Fig. 3) that are not covered by later figures. **a** Compositional mapping (where available) using CRISM data (colorizing CTX image F17_042328_1984_XN_18N281W). Green, blue, and red respectively trace pyroxenes, hydrated minerals, and Fe/Mg-phyllosilicates and/or carbonates using the LCPINDEX2, BD1900R2, and D2300 spectral parameters[82]. Boxes outline subsequent figures. **b** Earth's Mt. Sidley (126.1°W, 77.0°S) as viewed by Copernicus Sentinel-2 on 20 October 2020. Scale is the same as in (**a**). **c** Summit crater of Jezero Mons, with near-linear segments annotated (with dashed lines where they occur in the crater interior; HiRISE ESP_042328_1985_MIRB overlain on CTX image from (**a**)).

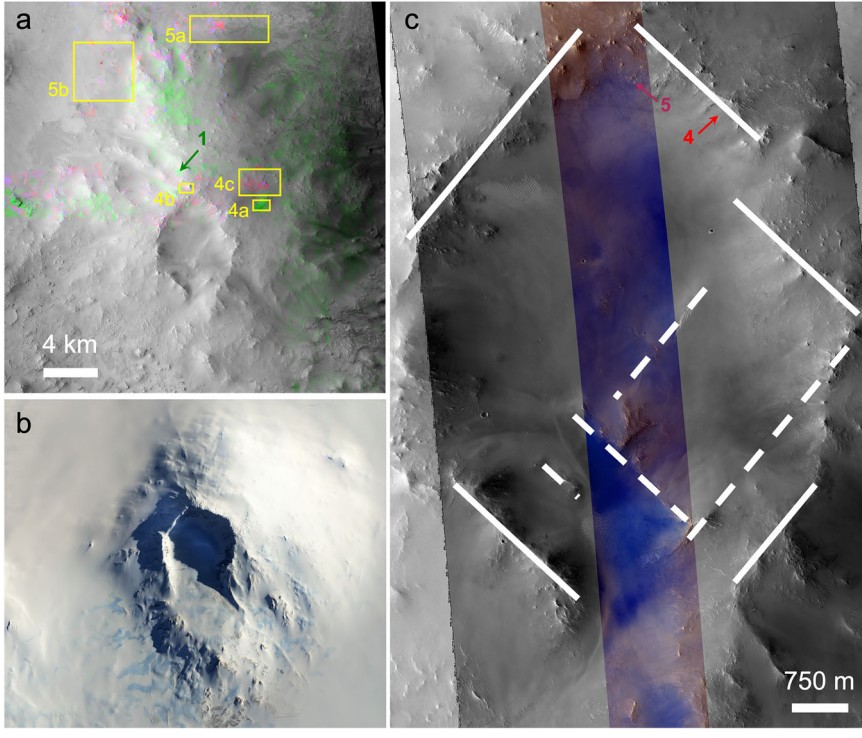

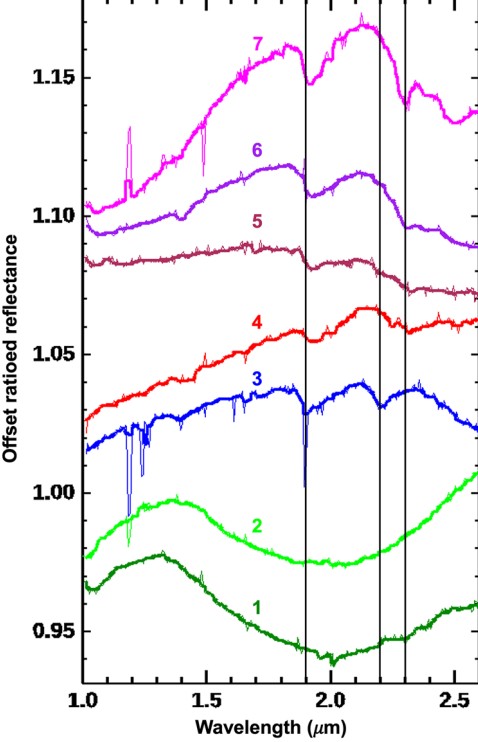

**Fig. 3 | Spectra from Jezero Mons.** #1 and 2 are consistent with pyroxenes and others with hydrated minerals as discussed in the text. #1 is from CRISM FRS0002AF61 and all others are from FRS00038C02, ratioed to nearby spectrally neutral areas (see Methods) and vertically offset for clarity. A five-channel median filter was applied (bold curves) to mitigate noise spikes (thin curves). Vertical lines are at 1.9, 2.2, 2.3 μm.

possibly layered pyroxene-bearing materials associated with the mountain extend overtop lighter bluish, polygonally fractured outcrops containing olivine and carbonate, both outside (Fig. 5a) and on the floor of Jezero crater (Fig. 5b, c). These flank materials within Jezero were previously grouped

with other crater rim and wall materials[13], but the marginal textures seen in Fig. 5 appear unique to the southeast side of the crater adjacent to Jezero Mons.

The margins of Jezero Mons superposing the olivine/carbonate-bearing unit both in and outside Jezero crater imply that the mountain at least partly postdates this unit, but by how much? With a surface area ~600 km², Jezero Mons is intermediate between the landform scales deemed statistically viable for age estimation via crater counting (>1000 km²) versus those found to have large age uncertainties (~100 km² and smaller)[35], although this depends strongly on surface age and subsequent resurfacing rate[36]. The few superposed craters of diameter >400 m suggest a surface age of ~1.0 ± 0.4 Ga (Fig. 6), but the abundances of smaller craters deviate from Mars isochrons, implying surface degradation since its formation. In particular, the slope of the log-log cumulative crater abundance plot (Fig. 6) versus sizes <400 m is shallower by ~2 than an isochron slope, consistent with diffusional processes that act on unconsolidated materials (and inconsistent with steady exhumation of bedrock or passive dust mantling)[37]. This evidence for degradation and crater obliteration implies that the inferred age should be considered a lower limit, and that current surface properties (e.g., thermal inertia) could also be influenced by this degradation.

To facilitate evaluation of Jezero Mons as a potential volcano, we measured its morphometry and that of similarly sized, previously recognized volcanoes on Earth and Mars using identical methods (Table 1), following a long tradition of comparative planetary volcano morphometry[38,39]. Most of the individually named martian shield volcanoes are an order of magnitude larger than Jezero Mons[40,41], but two are within a factor of 3, Jovis and Uranius Tholi, although embayment by younger lava flows has partially buried their flanks[42]. A similarly sized mountain with summit crater in Thaumasia Planum has been interpreted as an explosive volcano[43], and two of the first suggested martian composite volcanoes (i.e., stratovolcanoes)—Zephyria and Apollinarus Tholi[44]—are even more similar in size to Jezero Mons (Table 1). For a terrestrial comparison, we measured Antarctica's Mt. Sidley (Fig. 2b), which was recently cited as an analog for a larger suggested volcanic cone in Mars's Argyre basin[45] but is more similar in size to Jezero Mons (Table 1).

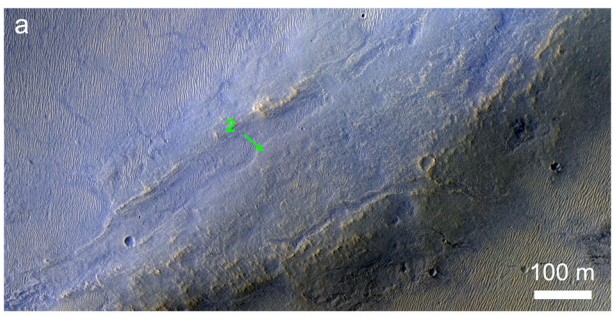

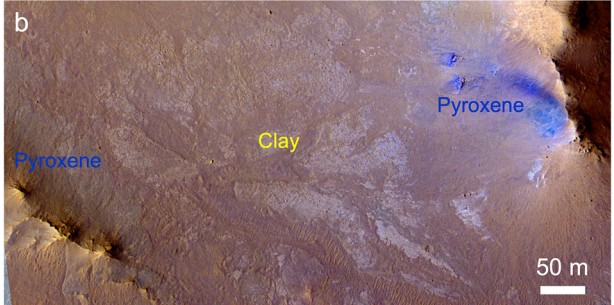

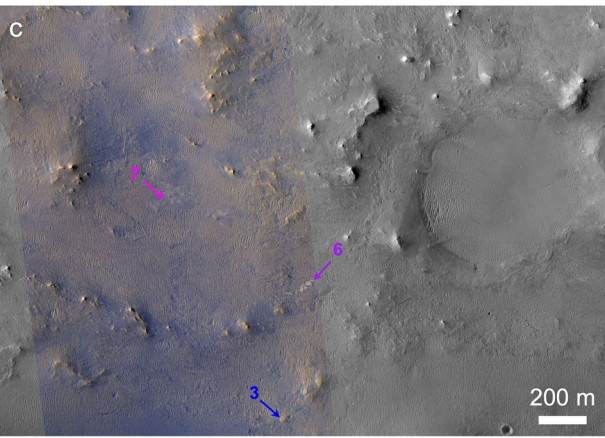

**Fig. 4 | Surface textures of Jezero Mons.** Arrows point to locations of CRISM spectra (Fig. 3). **a** Pyroxene-rich ridge east of the summit crater (HiRISE ESP_069719_1985). **b** Pyroxene-rich ridges and phyllosilicate-bearing materials just north of the summit crater (HiRISE ESP_033150_1985). **c** Hydrated minerals surrounding a quasi-circular filled depression on the northeast flank (ESP_069719_1985). All panels show Merged IRB products[84] to display color where available (each stretched to enhance contrast) and otherwise RED data in grayscale.

## Discussion

The gross morphometry of Jezero Mons is very similar to previously identified composite volcanoes, with the same aspect ratio (height/diameter) as Zephyria and Apollinarus Tholi (and Earth's Mt. Sidley, which has close to the modal ratio for terrestrial composite volcanoes[46]) as well as average flank slopes and a proportionate summit crater size (details in Methods) that are equivalent within measurement error. The summit crater's polygonal shape is reminiscent of structural control observed for other hypothesized martian composite volcano craters[47]. Twenty other mountains of similar height, width, and slopes to Jezero Mons in Mars's south polar region have also been interpreted as volcanic[48,49], although their detailed morphology and mineralogy are different due to likely eruption under regional ice sheets. Zephyria, Apollinarus, and the interpreted volcano in Thaumasia also have lower thermal inertias than their surrounding regions, with values within 20% of Jezero Mons (Table 1). As previously argued for these other edifices[43], low thermal inertia on Jezero Mons is consistent with a surface dominated by fine-grained particles such as ash. By contrast, shield volcano Uranius Tholus has a higher apparent thermal inertia than its

surroundings, consistent with near-surface bedrock (lava flows). Although Uranius's aspect ratio and flank slopes are within a factor of 2 of Jezero Mons, aspect ratio is smaller for most shield volcanoes (e.g., only 0.018 for Jovis Tholus). In particular, among hundreds of shield volcanoes mapped in the Tharsis region with diameters bracketing that of Jezero Mons[50,51], none match its height and all have lower ratios of crater width to basal diameter, typically by an order of magnitude. In combination with its poor crater retention, these factors argue that if Jezero Mons is a volcano, then it is more likely a composite volcano than an effusive shield[52,53].

Alternatively, mud volcanism can produce landforms that superficially resemble some igneous volcanoes. Morphometric comparisons can help to distinguish between these eruptive mechanisms[54]; in the case of Jezero Mons, it is an order of magnitude larger than even the largest subaerial mud volcanoes on Earth[55]. It has been argued on physical grounds that martian mud volcanoes should be morphometrically similar to those on Earth[56], weakening a mud volcano hypothesis for Jezero Mons.

If Jezero Mons is not a volcano, then its summit crater must presumably be of impact origin despite its non-circularity. Elsewhere on Mars, craters elevated on pedestals are interpreted to have armored the surrounding surface against erosion as part of the impact process[57,58]. However, no pedestal craters have previously been identified in this region[58], and where they do occur, the crater typically sits on a nearly planar surface bounded by steep scarps, unlike the gradual slopes observed on Jezero Mons's flanks (Fig. 1c). Furthermore, the low thermal inertia and low crater density on Jezero Mons suggest that it is less resistant to erosion than its surroundings, rather than more resistant as expected for a pedestal crater. The mountain could instead be an ancient massif, many of which have been mapped elsewhere around the Isidis basin[59], but its distinctly low thermal inertia and crater counts argue against this as well (Table 2).

The minerals that we identified on Jezero Mons are broadly similar to those mapped from orbit elsewhere around Jezero crater, which is consistent with the summit crater having exposed them via impact. But the detection of pyroxene(s) is also consistent with the volcanic hypothesis, and the broad absorption centered near 2.0 μm (Fig. 3) is more similar to that found on Jezero crater's floor than to the ~1.8 μm band center found on other parts of the crater rim[14]. The secondary minerals we identified cannot be primary products of volcanism; some of these could have been excavated by impact from older crust predating Jezero Mons (Fig. 4c), but hydrated minerals in and around the summit crater suggest more recent alteration (if this crater was alternatively formed by impact, it would be too small to achieve sufficient hydrothermalism for substantial mineralogic alteration[60]). Clay minerals also occur on and near the summits of the interpreted explosive volcanoes in Thaumasia Planum[43], where it was argued that the fine-grained, glassy nature of pyroclastic materials could have made them especially susceptible to alteration. Similar processes could have occurred at Jezero Mons, where the spectral evidence for partial chloritization of summit clays would be consistent with elevated temperatures[34] from volcanic activity. Opaline silica has also previously been observed in association with a volcanic cone on nearby Syrtis Major[61], but this mineral can form in a range of environments[12,33].

If Jezero Mons is volcanic, then the southern breach in its summit crater could have formed via sector collapse, analogous to Mt. Sidley (Fig. 2) or the 1980 Mt. St. Helens event[62,63]. Two other erosional troughs on the mountain's northwest flank appear to have sourced lobate deposits that extend into Jezero crater (Fig. 5b, c); these look smooth at meter scales, not unprecedented for martian lavas[64], but could instead be interpreted as lahar, debris flow or pyroclastic flow deposits. Whether flows from Jezero Mons could have extended farther west into Jezero crater than those clearly visible today depends on unknown factors including rheology and the crater's paleo-topography. But in any case, modeling suggests that even the thin modern atmosphere could disperse ash up to ~50 km[65], exactly the range needed to distribute particles from Jezero Mons across the full Jezero crater floor. Other topographic prominences within ~100 km of Jezero could be part of a related volcanic field[66], but Jezero Mons is the largest and least ambiguous of these.

**Fig. 5 | Margins of Jezero Mons. a** Northern margin (arrows) on the plains outside Jezero crater's eastern rim (CaSSIS MY36_021299_017_1).
**b** Northwestern margin on the southeast floor of Jezero crater (CaSSIS MY37_026393_165_0).
**c** Close-up view of possible eroded flow edge (arrows) and underlying lighter-toned crater floor materials with aeolian ripples superposed (HiRISE ESP_021678_1985).

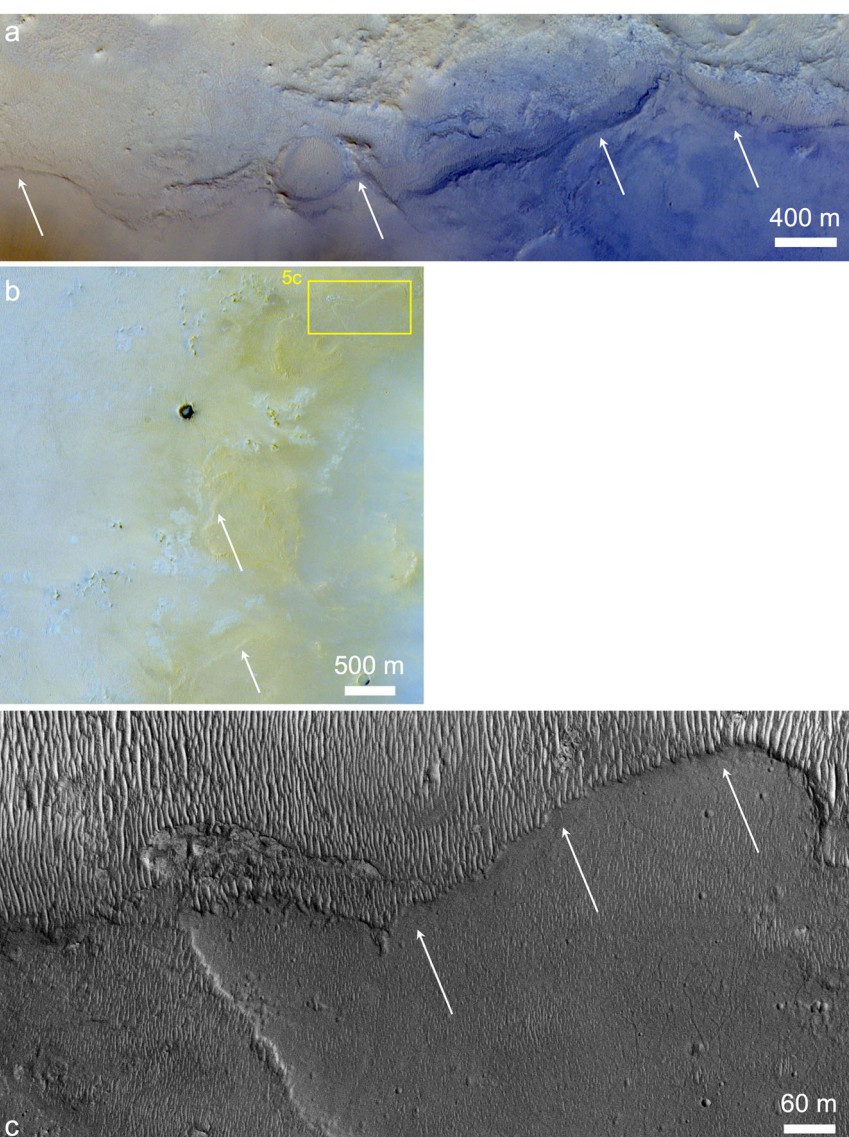

Of the rocks explored by Perseverance to date, the olivine-rich Séítah formation appears unlikely related to Jezero Mons, given their different orbital spectral signatures and the likely greater age (~3.82 Ga) of Séítah[67]. A pyroclastic origin for the olivine- and carbonate-bearing unit in and around Jezero crater was suggested based on orbital observations[68,69] and may be correct for a subset of it, but the cumulate textures found in situ argue that intrusive and/or low-viscosity effusive components are also present, perhaps all from related magmas[19,21]. In any case, if Séítah is part of this ~70,000 km$^2$ regional unit[68] (as suggested by compositional similarities between Jezero delta sediments and Séítah, consistent with the latter being part of the regional unit also present within the delta's watershed[70]), then Jezero Mons could not be the only source.

The darker-toned unit on Jezero's floor (Máaz formation) has recently been suggested to be part of another regionally widespread (~50,000 km$^2$) pyroclastic-dominated unit[71,72], but this has yet to be confirmed in situ[26]. This unit within Jezero was first estimated to be of ~1.4 Ga age[5], potentially consistent with the young surface age measured for Jezero Mons, but subsequent estimates of the crater floor unit's age have ranged from ~2.3 Ga[23] to ~3.45 Ga[13], either younger[5,13] or older than Jezero's deltaic deposits[24,71]. These diverging interpretations can be reconciled if Máaz formation emplacement began >3.5 Ga, likely after the first fluvial activity in Jezero[16] but before the final delta deposits[14,24], and then continued after the last fluvial

activity[66], with time-varying physical properties or subsequent burial and exhumation varying across the crater to produce a heterogeneous cratering record[73]. This complex history combined with large uncertainties on the age of Jezero Mons (beyond the clear superposition of some flows from the edifice overtop Séítah-equivalent rocks) allows for the possibility that Jezero Mons could have contributed to the Máaz formation. Indeed, the mixture of low- and high-calcium pyroxenes inferred from our spectra is similar to that found in situ in the Máaz formation[25,74], where secondary minerals including Fe/Mg-phyllosilicates were also observed[74,75].

Confirmation of any direct relationship between Jezero Mons and the rocks of western Jezero crater may ultimately require closer inspection of the former than can be achieved from orbit. Regardless, the fact that such a well-exposed potential volcanic edifice could have gone unrecognized for so long at one of the best-studied sites on Mars suggests that many more ancient volcanoes may yet await identification across the planet[47,76,77].

## Methods
### Thermal inertia
Thermal inertia (TI) of a planetary surface quantifies its resistance to temperature changes and is a product of thermal conductivity, material density, and heat capacity. Higher TI typically indicates coarser sizes of loose particles, induration, and/or exposure of bedrock[78]. It can be deduced

from measurements of nighttime infrared brightness, such as those by the Thermal Emission Imaging System (THEMIS) on NASA's Mars Odyssey spacecraft, which provides the highest spatial resolution available (~100 m/pixel), with estimated ~20% accuracy[79]. We computed average values and standard deviations across each martian edifice in Table 1 using

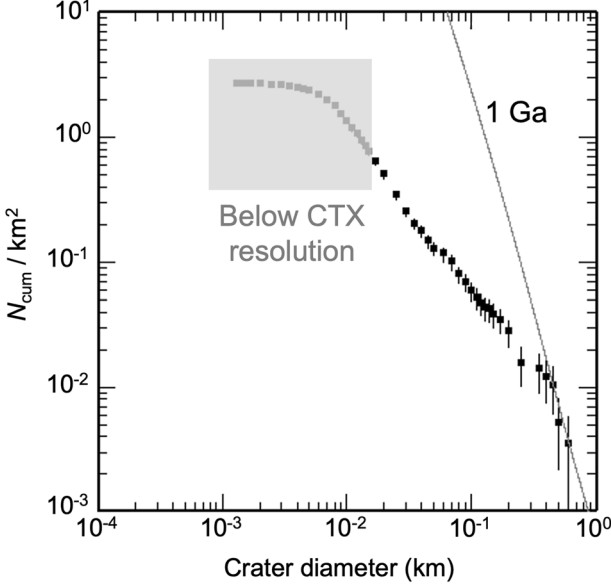

**Fig. 6 | Crater size-frequency distribution for flanks of Jezero Mons.** We counted 1538 craters of inferred impact origin (excluding the summit crater) across a total area ~584 km². Statistical error bars computed by CraterTools[86] allow surface ages of ~1.0 ± 0.4 Ga to fit the largest ($D \gtrsim 400$ m) superposed craters.

THEMIS thermal inertia (TIN) products with an image rating of at least 4 (on a 1–7 quality scale defined by the THEMIS team based on consideration of exposure, missing lines, instrument noise, and atmospheric features). Boundaries for each mountain were delineated based on examination of visible and thermal images, resulting in measured surface areas of 584, 661, 574, and 470 km² for the Jezero, Zephyria, Apollinaris, and Thaumasia edifices, respectively. Surficial materials may contribute to these thermal inertias, but we note that all values measured were approximately twice that observed by THEMIS for uniformly dust-blanketed regions (~100 J m⁻² K⁻¹ s⁻¹/² or less[79]).

### Hyperspectral mapping

To infer surface compositions, the highest spatial resolution for hyperspectral mapping is provided by the Compact Reconnaissance Imaging Spectrometer for Mars (CRISM) on NASA's Mars Reconnaissance Orbiter (MRO). CRISM has two detectors spanning the visible/near-infrared (~0.4–1.0 μm) and infrared wavelengths (~1.0–3.9 μm), respectively[80]. Two CRISM observations at half resolution (~40 m/pixel) cover the northern flank of Jezero Mons (HRL00010963 and HRL000116C6); for these we analyzed CRISM Targeted Empirical Records (TERs), data products that combine both wavelength ranges and include systematic corrections for atmospheric gas absorptions, aerosol scattering, photometric effects, and instrument artifacts[81]. Two additional CRISM images cover smaller areas around the summit of Jezero Mons (FRS0002AF61 and FRS00038C02), and another covers part of its western flank (FRS000281D1), at full resolution (~20 m/pixel); as of this writing, TERs are not yet available for these newer observations, so we instead used Targeted Reduced Data Record (TRDR) files[80], focusing on the infrared wavelengths (Fig. 3), which are most diagnostic of mineralogy. For the TRDR files we performed standard photometric and atmospheric corrections: respectively, dividing I/F by the cosine of the solar incidence angle and by a scaled atmospheric transmission spectrum derived from observations over Olympus Mons[80]. For both

---

**Table 1 | Morphometry and thermophysical properties of Jezero Mons compared to previously identified volcanoes**

|  | Jezero Mons | Zephyria Tholus[44] | Apollinaris Tholus[44] | Thaumasia "Volcano 1"[43] | Uranius Tholus[42] | Mt. Sidley[62] |
|---|---|---|---|---|---|---|
| Diameter (km) | 20.9 | 28.7 | 26.8 | 71.0 | 61.6 | 18.9 |
| Crater width (km) | 6.2 | 8.0 | 7.7 | 21.5 | 20.8 | 4.7 |
| Height (km) | 1.57 | 2.18 | 2.00 | 1.95 | 2.85 | 1.42 |
| Crater width/ diameter ratio | 0.31 | 0.28 | 0.29 | 0.30 | 0.34 | 0.25 |
| Height/diameter ratio | 0.076 | 0.076 | 0.075 | 0.027 | 0.046 | 0.076 |
| Thermal inertia (J m⁻² K⁻¹ s⁻¹/²) | 239 ± 21 | 208 ± 25 | 196 ± 31 | 198 ± 22 | N/Aᵃ | N/A |
| Flank slope (degrees) | 8.2 | 9.2 | 8.6 | 2.5 | 6.9 | 5.9 |

ᵃNo THEMIS thermal inertia (TIN) products available with image rating 4+; however, nighttime IR brightness suggests Uranius Tholus has higher TI than its surroundings. Its measured diameter and height should also be considered minimum estimates given evidence for burial by younger lavas[42].

---

**Table 2 | Summary of hypotheses vs. observations of Jezero Mons**

|  | Size & shape | Low thermal inertia | Mafic minerals | Clay minerals | Less cratered than surroundings |
|---|---|---|---|---|---|
| Ancient massif w/ coincidental impact crater | ? | X | ✓ | ✓ | X |
| Pedestal crater | X | X | ✓ | ✓ | X |
| Mud volcano | X | ✓ | ? | ✓ | ✓ |
| Shield volcano | ? | X | ✓ | X | ? |
| Composite volcano | ✓ | ? | ✓ | ? | ? |

Size and shape (morphometry) calculated from MOLA-HRSC data, thermal inertia from THEMIS, mineralogy from CRISM, and crater counts primarily from CTX (see Methods). A check mark indicates the observation appears consistent with that hypothesis, question mark denotes ambiguity, and X signifies inconsistency with what that hypothesis predicts.

---

observation types, spectra averaged from pixels of interest were divided by spectral averages from dusty or otherwise spectrally neutral regions observed in the same CRISM detector columns, producing the ratioed reflectance spectra in Fig. 3. This spectral ratio method reduces residual artifacts from atmospheric removal and from the CRISM instrument itself, and has been standard across most prior publications of CRISM data.

Areas of interest for plotting spectra were identified with the help of spectral summary parameters, which quantify absorption band depths or other spectral characteristics on a pixel-by-pixel basis[82]. These parameters enable mapping of the presence and relative spectral prominence (likely related to abundance and grain size[83]) of minerals such as pyroxenes and hydrated phyllosilicates or carbonates (Fig. 2a). To expand our mineral mapping coverage of Jezero Mons beyond where CRISM targeted observations were acquired, we used the Multispectral Reduced Data Record (MRDR) tile (#1250) covering this region, which is a mosaic of CRISM multispectral (72-wavelength) survey data at reduced resolution of ~200 m/pixel[80]. In this case, the MRDR data added coverage of the far eastern flank of Jezero Mons.

### Morphology and morphometry

The finest morphological details resolvable from orbit—about an order of magnitude finer than is achievable by Perseverance's Mastcam-Z (Fig. 1c) or SuperCam-RMI cameras, given the rover's current and likely future distance from Jezero Mons—are provided by the ~30 cm/pixel images of MRO's High Resolution Imaging Science Experiment (HiRISE). HiRISE coverage of Jezero Mons is comparably (in)complete to that by CRISM, and color HiRISE coverage is even more limited, spanning only the central 20% of each HiRISE image[84]. Where available, we used contrast-enhanced HiRISE IRB color products (Fig. 4), which display the IR, RED, and BG filter images in red, green, and blue, respectively[84]. Some additional color coverage is provided by the Colour and Stereo Surface Imaging System (CaSSIS) onboard ESA's ExoMars Trace Gas Orbiter, which includes four color filters and images at ~4.6 m/pixel[85]. The CaSSIS images in Fig. 5 display the NIR, PAN, and BLU filter images in red, green, and blue, respectively.

The Context camera (CTX) on MRO provides complete coverage of Jezero Mons in a single panchromatic band, so we primarily used its images for counting superposed impact craters, supplementing with HiRISE imagery where available to confirm and identify smaller craters but noting which were too small to resolve with CTX (Fig. 6), for which our counts are therefore incomplete. We used CraterTools[86] in ArcMap to outline all craters, compute statistics and find the best-fit age isochron.

We used a meter-scale digital terrain model (DTM) produced from a stereo pair of HiRISE images (ESP_033150_1985 and ESP_042328_1985) by the photogrammetry lab at the HiRISE Operations Center (University of Arizona) covering most of Jezero Mons's summit crater to examine fine-scale structure within the crater (Fig. 2c). However, for measuring other martian volcanoes, the best universally available topographic dataset was the Mars Orbiter Laser Altimeter (MOLA) – High Resolution Stereo Camera (HRSC) blended global mosaic[87], which we therefore used for all morphometry in Table 1. Specifically, the diameter, crater width, height, and ratios between these were measured four times for each edifice—delineated manually based on breaks in slope[88] and on its distinctive properties relative to surrounding areas in visible and thermal images—using profiles oriented north-south, east-west, northeast-southwest, and northwest-southeast, respectively. The four values of each parameter were then averaged to produce the numbers in Table 1 (and for this reason, e.g., the average aspect ratio in the table is not necessarily exactly equal to the ratio of average height to average diameter). The average slope for each edifice was calculated from fifteen different profile lines spanning its flanks, equally angularly spaced, in order to produce a single representative number for direct comparison in spite of substantial azimuthal variations in some cases.

### Data availability

All data used for this work are publicly available. CRISM data and the software toolkit we used to analyze them can be downloaded from NASA's Planetary Data System (https://pds-geosciences.wustl.edu/missions/mro/crism.htm). HiRISE images and DTMs can be accessed at (https://www.uahirise.org). CaSSIS images are available at (https://observations.cassis.unibe.ch). Other planetary datasets and visualization capabilities are available within the free JMARS software provided by Arizona State University (https://jmars.asu.edu).

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

## Acknowledgements

S.C.C.Q. was supported by Georgia Tech's 2021 Research Experience for Undergraduates program sponsored by 3M corporation. J.J.W. was supported by NASA funding for Co-Investigators on HiRISE and CaSSIS. CaSSIS is a project of the University of Bern and funded through the Swiss Space Office via ESA's PRODEX program. The instrument hardware development was also supported by the Italian Space Agency (ASI) (ASI-INAF agreement 2020-17-HH.0), INAF/Astronomical Observatory of Padova, and the Space Research Center (CBK) in Warsaw. Support from SGF (Budapest), the University of Arizona (Lunar and Planetary Lab), and NASA are also gratefully acknowledged. Operation support from the UK Space Agency under grant ST/R003025/1 is also acknowledged. We thank Emmy Hughes and Steve Ruff for illuminating conversations, and Kris Akers, Sarah Sutton, and Matt Chojnacki for efficient HiRISE DTM production. Feedback from David Crown, Ernst Hauber, and two anonymous reviewers improved the paper substantially. We are grateful to the IAU Working Group for Planetary System Nomenclature for their timely approval of our name request for Jezero Mons.

## Author contributions

S.C.C.Q. led the compilation and analysis of THEMIS, MOLA/HRSC, and all MRO datasets. J.J.W. conceived the project, targeted Jezero Mons with orbital imagers since 2007, and assisted with data analysis. F.R.H. co-advised the project, generated digital terrain models, and provided hardware and software. J.B.A. assisted with thermal inertia interpretation and figure generation. J.J.W. wrote the paper with significant input from all other authors.

## Competing interests

The authors declare no competing interests.
