## [Transparent Peer Review file · Communications Earth & Environment]

Evidence for a composite volcano on the rim of Jezero crater on Mars

Corresponding Author: Professor James Wray

This manuscript has been previously reviewed at another Nature Portfolio journal. This document only contains reviewer comments and rebuttal letters for versions considered at Communications Earth & Environment.

This file contains all editorial decision letters in order by version, followed by all author rebuttals in order by version. Attachments originally included by the reviewers as part of their assessment can be found at the end of this file.

Version 0:

Decision Letter:

Dear Professor Wray,

Please allow me to sincerely apologise for the long delay in sending a decision on your manuscript titled "Evidence for a stratovolcano on the rim of Jezero crater on Mars". This was due to the fact that neither of the previous reviewers were able to send further reports. However, your manuscript has now been seen by 2 replacement reviewers, and we include their comments at the end of this message. They find your work of interest, but some important points are raised. We are interested in the possibility of publishing your study in Communications Earth & Environment, but would like to consider your responses to these concerns and assess a revised manuscript before we make a final decision on publication.

We therefore invite you to revise and resubmit your manuscript, along with a point-by-point response that takes into account the points raised. Please highlight all changes in the manuscript text file.

Please use the following link to submit your revised manuscript, point-by-point response to the referees' comments (which should be in a separate document to any cover letter), a tracked-changes version of the manuscript (as a PDF file) and the completed checklist:

Link Redacted

We hope to receive your revised paper within six weeks; please let us know if you aren't able to submit it within this time so that we can discuss how best to proceed. If we don't hear from you, and the revision process takes significantly longer, we may close your file. In this event, we will still be happy to reconsider your paper at a later date, as long as nothing similar has been accepted for publication at Communications Earth & Environment or published elsewhere in the meantime.

Please do not hesitate to contact us if you have any questions or would like to discuss these revisions further. We look forward to seeing the revised manuscript and thank you for the opportunity to review your work.

Best regards,

Joe Aslin

Deputy Editor,
Communications Earth & Environment
<https://www.nature.com/commsenv/>
Twitter: @CommsEarth

EDITORIAL POLICIES AND FORMATTING

Editorial Policy: [Policy requirements](https://www.nature.com/documents/nr-editorial-policy-checklist.pdf) (Download the link to your computer as a PDF.)

- Behavioural and social science
- Ecological, evolutionary & environmental sciences
- Life sciences

<https://www.nature.com/documents/nr-reporting-summary.zip>

Furthermore, please align your manuscript with our format requirements, which are summarized on the following checklist: [Communications Earth & Environment formatting checklist](https://www.nature.com/documents/commsj-phys-style-formatting-checklist-article.pdf)

and also in our style and formatting guide [Communications Earth & Environment formatting guide](https://www.nature.com/documents/commsj-phys-style-formatting-guide-accept.pdf).

*** DATA: Communications Earth & Environment endorses the principles of the Enabling FAIR data project (<http://www.copdess.org/enabling-fair-data-project/>). We ask authors to make the data that support their conclusions available in permanent, publically accessible data repositories. (Please contact the editor if you are unable to make your data available).

All Communications Earth & Environment manuscripts must include a section titled "Data Availability" at the end of the Methods section or main text (if no Methods). More information on this policy, is available at <http://www.nature.com/authors/policies/data/data-availability-statements-data-citations.pdf>.

If a community resource is unavailable, data can be submitted to generalist repositories such as [figshare](https://figshare.com/) or [Dryad Digital Repository](http://data.dryad.org/). Please provide a unique identifier for the data (for example a DOI or a permanent URL) in the data availability statement, if possible. If the repository does not provide identifiers, we encourage authors to supply the search terms that will return the data. For data that have been obtained from publically available sources, please provide a URL and the specific data product name in the data availability statement. Data with a DOI should be further cited in the methods reference section.

Please refer to our data policies at

<http://www.nature.com/authors/policies/availability.html>.

REVIEWER COMMENTS:

Reviewer #3 (Remarks to the Author):

I have had the opportunity to read the rebuttal and the manuscript of entitled "Evidence for a stratovolcano on the rim of Jezero crater on Mars" and assess how well its authors managed to answer the raised questions by a couple of reviewers. Namely I was asked to especially focus about rebuttal for comments of reviewer 1.

I think that the rebuttal and updated version of the manuscript dealt with the objections adequately and that none of the critical comments remained unresolved. From my point of view, the manuscript meets the requirements to be accepted for publication by the editor. It contains interesting ideas, which will inspire further scientific debate and hence represent a step

forward in our aim to understand the evolution of Mars.

However, I have two minor comments for the authors to consider.

1) The term stratovolcano is obsolete and the terrestrial volcanologist's community is preferring to use the term "composite volcano" as this type of volcanoes do not have to have regularly alternating layers which the term "strata" imply. Therefore I would propose to replace this term and use the "modern" one.

2) Actually from those who are not experts on martian geography, it is hard to understand from Figure 1, where is the position of Jezero Mons. Both from global and regional perspective. The figure would benefit if a small MOLA based globe is added and the position of the Jezero crater/Mons is marked, and if the Jezero crater itself is shown in nadir/CTX mosaic image is shown with the well-known delta and the position of Jezero Mons is marked there. This would help to make the figure/article more accessible for volcanologists, which are not working on Mars.

Reviewer #4 (Remarks to the Author):

see review in attached file

** Visit Nature Research's author and referees' website at <http://www.nature.com/authors> for information about policies, services and author benefits **

Communications Earth & Environment is committed to improving transparency in authorship. As part of our efforts in this direction, we are now requesting that all authors identified as 'corresponding author' create and link their Open Researcher and Contributor Identifier (ORCID) with their account on the Manuscript Tracking System prior to acceptance. ORCID helps the scientific community achieve unambiguous attribution of all scholarly contributions. You can create and link your ORCID from the home page of the Manuscript Tracking System by clicking on 'Modify my Springer Nature account' and following the instructions in the link below. Please also inform all co-authors that they can add their ORCIDs to their accounts and that they must do so prior to acceptance.

Version 1:

Decision Letter:

Dear Professor Wray,

Your manuscript titled "Evidence for a composite volcano on the rim of Jezero crater on Mars" has now been seen by our reviewers, whose comments appear below. In light of their advice we are delighted to say that we are happy, in principle, to publish a suitably revised version in Communications Earth & Environment.

We therefore invite you to revise your paper one last time to address the remaining concerns of our reviewers. At the same time we ask that you edit your manuscript to comply with our format requirements and to maximise the accessibility and therefore the impact of your work.

EDITORIAL REQUESTS:

****Please take care to match our formatting and policy requirements. We will check revised manuscript and return manuscripts that do not comply. Such requests will lead to delays. ****

SUBMISSION INFORMATION:

OPEN ACCESS:

Communications Earth & Environment is a fully open access journal. Articles are made freely accessible on publication. For further information about article processing charges, open access funding, and advice and support from Nature Research, please visit <https://www.nature.com/commsenv/open-access>

Link Redacted

**This url links to your confidential home page and associated information about manuscripts you may have submitted or be reviewing for us. If you wish to forward this email to co-authors, please delete the link to your homepage first **

Best regards,

Joe Aslin

Deputy Editor,
Communications Earth & Environment

Consulting Editor,
Communications Sustainability

<https://www.nature.com/commsenv/>
Twitter: @CommsEarth

REVIEWERS' COMMENTS:

Reviewer #3 (Remarks to the Author):

No further comments on my part. I can recommend the article to the editor for possible publication.

Reviewer #4 (Remarks to the Author):

Reviewer 4 (David A. Crown)

I have reviewed the revisions made to the manuscript and the authors' responses to the previous review comments. The authors have done a thorough and commendable job of responding and refining their manuscript accordingly. I offer a few comments below for the authors' consideration prior to publication.

Lines 62-71 (original comment and line numbers): I appreciate the explanation offered and revisions made in response. I recommend one or both of the following re this point:

- a) Include in the Results section the substance of the review comment response in the manuscript to explain that the definition of the outer margin of Jezero Mons is not clear and why.
- b) Add a dashed line to Figure 1c to indicate the approximate boundary of the volcano with "?" inserted where the boundary is most uncertain.

Lines 126-137 and Table 1: I think you should refer the reader to the Methods section re the morphometric parameters. The explanation offered is reasonable but I would still prefer to see even a rough outline of its base (as suggested above).

Table 1. Should there be citation noted for Uranus Tholus and Mt. Sidley on this table?

Table 2: I think you should probably define what the X's, ?'s, and boxes mean in a footnote.

Figure 1:

a) Include color scale bar for Fig 1d.

b) The topographic profile is a nice addition. It would be useful to give an azimuth for the topographic profile or show the start and end points on Fig. 1e.

Figure 5: Consider adding arrows to Figs 5a and 5b to show features of interest.

** Visit Nature Research's author and referees' website at <http://www.nature.com/authors> for information about policies, services and author benefits **

David A. Crown
Planetary Science Institute
June 2, 2024

General Comments:

In providing review comments for this article, I have prepared my own review as well as read through the comments from Reviewers 1 and 2 from the initial review stage (redacted). I was asked to specifically comment on the authors' responses to Reviewer 2. Note: I did not have access to the annotated manuscript submitted by Reviewer 2 or the original submitted version of the manuscript. The authors' responses to review comments by both Reviewers 1 and 2 from the initial review stage seem for the most part to be thorough and appropriate. I have provided my own review below and also referred back to initial stage review comments re select issues.

I do find Jezero Mons of significant interest to the emerging story of the Jezero crater region and believe the lines of evidence discussed in this manuscript to be worthy of pursuit and publication. There are some parts of the manuscript that I believe would benefit from further clarification and tightening up of their presentation. The manuscript makes a reasonable case for the interpretation of Jezero Mons as a stratovolcano (and not a mud volcano or pedestal crater). I find the most compelling lines of evidence to be its gross morphology and the presence of igneous minerals and alteration signatures. Given the complex geologic environment and obvious signs of landscape degradation, the thermal inertia values do not seem to me to be diagnostic. I believe the crater size-frequency distribution could be explored more fully to potentially provide additional context. More thorough treatment of Jezero Mons' lower flanks should be incorporated to develop more clear hypotheses about potential formation mechanisms (e.g., lava flows, pyroclastic flows, and/or debris flows).

Specific Comments:

Abstract

Line 20: Based on responses to the initial review stage, should "Flow margins" be qualified here to "Possible flow margins" or this wording changed more generally?

It seems to me that the position of Jezero Mons right on the crater rim and your interpretation of it as a stratovolcano are the best evidence that it may have supplied materials to the interior of Jezero. The possible flow margins would represent another piece of evidence but it seemed to me that you were modifying this point based on comments by Reviewer 2. Please re-examine.

Introductory Sections (Lines 25-59): This is an excellent and concise summary that provides context for the manuscript.

Results

Lines 62-71: Given the complex geology of this region, it would be helpful to have at least a sketch map included as a figure showing the authors' interpretation of the outer margins of Jezero Mons (or superimpose this on Figure 1c). This is also relevant for defining the feature's flanks for slope calculations and making statements about surface morphology and thermophysical properties relative to surrounding terrains.

Lines 64-67: For context it would be useful to indicate the range of thermal inertia across Jezero Mons and any correlations between lows and highs in thermal inertia and surface features or compositional signatures.

Lines 72-77: It would be helpful to indicate datasets used for hyperspectral mapping (and/or refer the reader to the Methods section).

Lines 105-112: The constraints from crater counting are important. Please clarify whether the age constraint is from the full crater size-frequency distribution or a certain range of crater diameters that appears to best follow the isochron (> 400 m?). Please specify the fit range if appropriate. If you haven't already, you might plot this as a differential distribution. Examining the differential (instead of cumulative) distribution can be helpful in distinguishing formation (or surface stabilization) ages from resurfacing ages.

Lines 110-112: Your statement here about evidence for degradation and crater obliteration (which I think is accurate) suggests that your interpretations about thermal inertia reflecting possible volcanic materials of Jezero Mons should be qualified with this uncertainty. See also Lines 135-137. I don't view the thermal inertia as a strong constraint without more certainty that you are sampling the bedrock rather than surfaces dominated by particulate materials due to erosion/deposition.

Lines 113-123: See comment above about defining the margins of Jezero Mons. You need to define this to provide an understanding of your measurements of the mountain's size and shape. You should also cite previous studies of volcano morphometry here (and/or in the Methods section), including:

Crumpler, L.S., J.W. Head, and J.C. Aubele, 1996, Calderas on Mars: Characteristics, structure and associated flank deformation, in *Volcano Instability on the Earth and Other Planets* (W.J. McGuire, P. Jones, and J. Neuberg, eds.), *Geol. Soc. Spec. Publ.*, 110, 307–348.

Hodges, C.A., and H.J. Moore, 1994, *Atlas of Volcanic Landforms on Mars*, *U.S. Geol. Surv. Prof. Paper 1534*, 194 pp.

Pike, R.J., 1978, Volcanoes on the inner planets; some preliminary comparisons of gross topography, *Proc. Lunar Planet. Sci. Conf.*, 9th, 3239-3273.

Pike, R.J., and G.D. Clow, 1981, Revised classification of terrestrial volcanoes and a catalog of topographic dimensions with new results on edifice volume, *U.S. Geological Survey Open-File Report OF 81-1038*.

Plescia, J.B., 2004. Morphometric properties of Martian volcanoes, *J. Geophys. Res.*, 109, E03003, doi:10.1029/2002JE002031.

These papers on volcano morphometry may also be of interest:

Grosse, P., B. van Wyk de Vries, P.A. Euillades, M. Kervyn, and I.A. Petrinovic, 2012, Systematic morphometric characterization of volcanic edifices using digital elevation models, *Geomorphology*, 136, 114-131.

Grosse, P., P.A. Euillades, L.D. Euillades, et al., 2014, A global database of composite volcano morphometry, *Bull. Volcanol.* 76, 784.

Lines 113-123 and/or elsewhere: Regarding evaluating Jezero Mons as a potential stratovolcano, you might consider referencing one of the early studies that classified Martian volcanoes (e.g., Greeley and Spudis, 1981) and some recent review papers that treat explosive volcanism on Mars.

Broz, P., H. Bernhardt, S.J. Conway, and R. Parekh, 2021, An overview of explosive volcanism on Mars, *JVGR*, 409, doi.org/10.1016/j.volgeores.2020.107125.

Greeley, R. and P.D. Spudis, 1981, Volcanism on Mars, *Rev. Geophys. Space Phys.*, 19, 13-41.

Mouginis-Mark, P.J., Zimbelman, J.R., Crown, D.A., Wilson, L., and Gregg, T.K.P., 2022, Martian volcanism: Current state of knowledge and known unknowns, *Geochemistry*, doi.org/10.1016/j.chemer.2022.125886.

Line 144: Delete igneous. Seems redundant and you say “mud volcanism” in the next paragraph to differentiate.

Lines 178-179: Add references re sector collapse at Mt Sidley and Mt St Helens.

Lines 180-182 (See also comments on Figure 5): I think more complete treatment of the NW flank of Jezero Mons would be helpful here. I see two large valleys or erosional troughs on the NW flank, from which dark, lobate deposits extend into Jezero crater. I recommend you add debris flow to your list of possible interpretations.

Line 181: Again, here I see an inconsistency relative to your response to Reviewer 2. If you want to call these flows (which seems reasonable), just make a case for it relative to a deposit with an erosional margin. You should support your statement re the smoothness of the possible flows with references to the literature. There are many possible studies to choose from.

I disagree with the statement that these are unusually smooth for Martian lava flows; see this reference for treatment of rough (a'a-like) and smooth (pahoehoe-like) lava flows in southern Tharsis:

Crown, D.A., and Ramsey, M.S., 2017, Morphologic and thermophysical characteristics of lava flows southwest of Arsia Mons, Mars, *J. Volc. Geotherm. Res.*, 342, *Pattern to Process: Remotely Sensed Observations of Volcanic Deposits and Their Implications for Surface Processes*, 13-28, doi:10.1016/j.jvolgeores.2016.07.008.

Methods

Lines 221-232: Re thermal inertia: It would be helpful to report the range and standard deviation for the thermal inertia values in Table 1 to provide better context. Perhaps also include surface area in Table 1. This also raises the question of what boundaries you used for each of the

volcanoes—can you provide a simple explanation or references to someone else's mapped boundaries?

Lines 284-296: Re diameters and slope values: A bit more complete documentation would be helpful here. Most importantly, please include how each volcano's boundary was determined. Re slopes, why did you use 15 different profile lines (and how were these selected) instead of averaging slopes from the N-S, E-W, NW-SE, and NE-SW lines used for other parameters? Perhaps, "Slopes" in Table 1 should be "Flank Slopes" given method?

Table 2

I recommend you include Jezero Mons in the title of the table as I believe this table is site specific.

Re Stratovolcano entry for low thermal inertia. One could reasonably expect the particle/block size characteristics of a stratovolcano to be highly variable (in both space and time) due to the combination of explosive and effusive eruptions that may have formed it. Thus, I would not conclude that low thermal inertia is a reliable indicator of a stratovolcano.

In general Table 2 seems overly simplistic/generalized. Size and shape could be expressed in more specific terms. Thermal inertia is problematic given uncertainty regarding erosional history and potential deposition of mantling deposits. Re "Less cratered than surroundings": The local geology is quite complex so it would help to be more specific here.

Figure 1:

Figure 1c caption: Do you need to change wording re "flows into Jezero crater" to be consistent with other revisions made re this in response to Reviewer 2?

I think Figure 1a has value but would also like to see a topographic profile as recommended by Reviewer 1 so I recommend that addition.

Figure 2:

Figure 2a caption, Line 535: Change "arrows" to "arrow."

Figure 2c: This image is too zoomed in to really appreciate the potential linear segments.

Figure 3 caption: Since the spectra are numbered, they should be referred to using the numbers in the figure caption (instead of "bottom two").

Figure 4: The images are all HiRISE excerpts but have very different colors. Caption could be expanded to provide more complete explanation of these images. For example, Figure 4c looks like it includes both b/w and color HiRISE.

Figure 5: What I perceive to be the margins of Jezero Mons are quite interesting. I am not sure these three images are the best choices to show the nature of the margins of Jezero Mons and relationships with adjacent terrain. Consider showing additional HiRISE and/or CTX to more fully document the margins (e.g., the deposits within Jezero crater).

Please also state whether or not (or uncertain) you interpret the possible flow edge to be a primary lava flow front or an erosional margin in the text with appropriate supporting information. Do the margins of Jezero Mons have consistent morphologic characteristics around its perimeter?

Figure 6: Suggested change for heading to *Crater size-frequency distribution for flanks of Jezero Mons*. Should specify in caption if the age you report is from fitting the whole distribution or some limited crater size range. It would also be useful to indicate in the caption at what crater size the CSFD appears to deviate from the isochron as this provides some indication of surface degradation.

We thank the editor and the two reviewers for their valuable comments. Our responses are interspersed below, in italics; in all cases at least one related change has been made to the paper. The only other changes are updates to two references, which have now been published, and to the secondary affiliation of the first author, who has now moved on to graduate school. To facilitate comparison to our prior submission, we have not renumbered references in this version (despite adding some new ones) to all still appear in the order they are cited, but we can assist with this shuffling upon acceptance if needed.

Reviewer #3:

I have had the opportunity to read the rebuttal and the manuscript of entitled “Evidence for a stratovolcano on the rim of Jezero crater on Mars” and assess how well its authors managed to answer the raised questions by a couple of reviewers. Namely I was asked to especially focus about rebuttal for comments of reviewer 1.

I think that the rebuttal and updated version of the manuscript dealt with the objections adequately and that none of the critical comments remained unresolved. From my point of view, the manuscript meets the requirements to be accepted for publication by the editor. It contains interesting ideas, which will inspire further scientific debate and hence represent a step forward in our aim to understand the evolution of Mars.

We thank the reviewer for this positive assessment.

However, I have two minor comments for the authors to consider.

1) The term stratovolcano is obsolete and the terrestrial volcanologist’s community is preferring to use the term “composite volcano” as this type of volcanoes do not have to have regularly alternating layers which the term “strata” imply. Therefore I would propose to replace this term and use the “modern” one.

We are grateful for this guidance and happy to oblige. All instances of “stratovolcano” have now been replaced with “composite volcano,” including in the title; in the first such instance in the main text (where we are referencing prior work that used the term “stratovolcano”), we now say “composite volcanoes (i.e., stratovolcanoes)” to confirm that we view these as synonymous.

2) Actually from those who are not experts on martian geography, it is hard to understand from Figure 1, where is the position of Jezero Mons. Both from global and regional perspective. The figure would benefit if a small MOLA based globe is added and the position of the Jezero crater/ Mons is marked, and if the Jezero crater itself is shown in nadir/CTX mosaic image is shown with the well-known delta and the position of Jezero Mons is marked there. This would help to make the figure/article more accessible for volcanologists, which are not working on Mars.

Revised as suggested: the figure now includes additional panels showing the MOLA globe with Jezero’s position, and a nadir/CTX mosaic image showing Jezero Mons relative to the rover’s landing site next to the delta.

Reviewer #4 (David A. Crown):

General Comments:

In providing review comments for this article, I have prepared my own review as well as read through the comments from Reviewers 1 and 2 from the initial review stage[redacted]. I was asked to specifically comment on the authors' responses to Reviewer 2. Note: I did not have access to the annotated manuscript submitted by Reviewer 2 or the original submitted version of the manuscript. The authors' responses to review comments by both Reviewers 1 and 2 from the initial review stage seem for the most part to be thorough and appropriate. I have provided my own review below and also referred back to initial stage review comments re select issues.

I do find Jezero Mons of significant interest to the emerging story of the Jezero crater region and believe the lines of evidence discussed in this manuscript to be worthy of pursuit and publication. There are some parts of the manuscript that I believe would benefit from further clarification and tightening up of their presentation. The manuscript makes a reasonable case for the interpretation of Jezero Mons as a stratovolcano (and not a mud volcano or pedestal crater). I find the most compelling lines of evidence to be its gross morphology and the presence of igneous minerals and alteration signatures. Given the complex geologic environment and obvious signs of landscape degradation, the thermal inertia values do not seem to me to be diagnostic. I believe the crater size-frequency distribution could be explored more fully to potentially provide additional context. More thorough treatment of Jezero Mons' lower flanks should be incorporated to develop more clear hypotheses about potential formation mechanisms (e.g., lava flows, pyroclastic flows, and/or debris flows).

We appreciate the reviewer's encouragement and detailed comments to help improve the paper, most of which we have implemented as fully as possible, as described below.

Specific Comments:

Abstract

Line 20: Based on responses to the initial review stage, should "Flow margins" be qualified here to "Possible flow margins" or this wording changed more generally?

It seems to me that the position of Jezero Mons right on the crater rim and your interpretation of it as a stratovolcano are the best evidence that it may have supplied materials to the interior of Jezero. The possible flow margins would represent another piece of evidence but it seemed to me that you were modifying this point based on comments by Reviewer 2. Please re-examine.

Indeed, our prior omission of "possible" in this instance was inadvertent; we have now thusly qualified the wording.

Introductory Sections (Lines 25-59): This is an excellent and concise summary that provides context for the manuscript.

We thank the reviewer for this kind description.

Results

Lines 62-71: Given the complex geology of this region, it would be helpful to have at least a sketch map included as a figure showing the authors' interpretation of the outer margins of Jezero Mons (or superimpose this on Figure 1c). This is also relevant for defining the feature's flanks for slope calculations and making statements about surface morphology and thermophysical properties relative to surrounding terrains.

Despite our best efforts (since 2007, as noted in the "Author contributions" statement!) high-resolution coverage of Jezero Mons with low-opacity images of sufficient contrast to confidently map its margins remains incomplete, as many more images continue to instead target the other side of Jezero crater where Perseverance roams. Coverage is slowly improving, and we hope that a robust map of Jezero Mons and its margins will be made in the future by others with more experience in geomorphologic mapping. In the meantime, we appreciate the point about needing to specify how we delineated the landform in terms of our morphometric and thermophysical measurements, and have added some additional detail on that, as noted below.

Lines 64-67: For context it would be useful to indicate the range of thermal inertia across Jezero Mons and any correlations between lows and highs in thermal inertia and surface features or compositional signatures.

We have added the following text: "values range from 136 to 444 J m⁻² K⁻¹ s^{-1/2}. Higher inertias typically occur along topographic ridges, including around the rim of [the summit crater]."

Lines 72-77: It would be helpful to indicate datasets used for hyperspectral mapping (and/or refer the reader to the Methods section).

We have now added a reference to the Methods section near the start of this paragraph.

Lines 105-112: The constraints from crater counting are important. Please clarify whether the age constraint is from the full crater size-frequency distribution or a certain range of crater diameters that appears to best follow the isochron (> 400 m?). Please specify the fit range if appropriate. If you haven't already, you might plot this as a differential distribution. Examining the differential (instead of cumulative) distribution can be helpful in distinguishing formation (or surface stabilization) ages from resurfacing ages.

Indeed, the fit was based on crater diameters $\gtrsim 400$ m, now clarified both in the text and in the Figure 6 caption. We would prefer to keep the figure as a cumulative plot since this is the more common presentation (given our wide target audience), including in the Kite & Mayer reference that we cite for our analysis of the isochron slope at smaller crater diameters.

Lines 110-112: Your statement here about evidence for degradation and crater obliteration (which I think is accurate) suggests that your interpretations about thermal inertia reflecting possible volcanic materials of Jezero Mons should be qualified with this uncertainty. See also Lines 135-137. I don't view the thermal inertia as a strong constraint without more certainty that you are sampling the bedrock rather than surfaces dominated by particulate materials due to erosion/deposition.

These are fair points. We added a half-sentence at (what was) lines 110-112 to acknowledge “that current surface properties (e.g., thermal inertia) could also be influenced by this degradation.” What we said on lines 135-137 is just that the low measured thermal inertia “is consistent with” ash, which seems accurate, even though we agree with the reviewer that this is far from diagnostic. We view the thermal inertia of Jezero Mons less as a strong quantitative constraint on its specific nature and more as a property that qualitatively highlights the mountain’s distinctness from the surrounding terrain (see Figure 1)—unlike previous maps that included it as part of a widespread “massive basement unit”—thereby motivating our study to use every tool we can to try to infer what it might be.

Lines 113-123: See comment above about defining the margins of Jezero Mons. You need to define this to provide an understanding of your measurements of the mountain’s size and shape. You should also cite previous studies of volcano morphometry here (and/or in the Methods section), including:

Crumpler, L.S., J.W. Head, and J.C. Aubele, 1996, Calderas on Mars: Characteristics, structure and associated flank deformation, in *Volcano Instability on the Earth and Other Planets* (W.J. McGuire, P. Jones, and J. Neuberg, eds.), *Geol. Soc. Spec. Publ.*, 110, 307–348.

Hodges, C.A., and H.J. Moore, 1994, *Atlas of Volcanic Landforms on Mars*, U.S. Geol. Surv. Prof. Paper 1534, 194 pp.

Pike, R.J., 1978, Volcanoes on the inner planets; some preliminary comparisons of gross topography, *Proc. Lunar Planet. Sci. Conf.*, 9th, 3239-3273.

Pike, R.J., and G.D. Clow, 1981, Revised classification of terrestrial volcanoes and a catalog of topographic dimensions with new results on edifice volume, U.S. Geological Survey Open-File Report OF 81–1038.

Plescia, J.B., 2004. Morphometric properties of Martian volcanoes, *J. Geophys. Res.*, 109, E03003, doi:10.1029/2002JE002031.

These papers on volcano morphometry may also be of interest:

Grosse, P., B. van Wyk de Vries, P.A. Euillades, M. Kervyn, and I.A. Petrinovic, 2012, Systematic morphometric characterization of volcanic edifices using digital elevation models, *Geomorphology*, 136, 114-131.

Grosse, P., P.A. Euillades, L.D. Euillades, et al., 2014, A global database of composite volcano morphometry, *Bull. Volcanol.* 76, 784.

Lines 113-123 and/or elsewhere: Regarding evaluating Jezero Mons as a potential stratovolcano, you might consider referencing one of the early studies that classified Martian volcanoes (e.g., Greeley and Spudis, 1981) and some recent review papers that treat explosive volcanism on Mars.

Broz, P., H. Bernhardt, S.J. Conway, and R. Parekh, 2021, An overview of explosive volcanism on Mars, *JVGR*, 409, doi.org/10.1016/j.volgeores.2020.107125.

Greeley, R. and P.D. Spudis, 1981, Volcanism on Mars, *Rev. Geophys. Space Phys.*, 19, 13-41.

Mouginis-Mark, P.J., Zimbelman, J.R., Crown, D.A., Wilson, L., and Gregg, T.K.P., 2022, Martian volcanism: Current state of knowledge and known unknowns, *Geochemistry*, doi.org/10.1016/j.chemer.2022.125886.

We had initially submitted this paper to [redacted], which requests no more than 70 references per article; seeing no such limitation in the guidelines for Communications Earth & Environment, we have now added all of these references, with gratitude to the reviewer for newly drawing several of them to our attention. (One in particular we now cite in our expanded description of how we defined the margins of Jezero Mons.)

Line 144: Delete igneous. Seems redundant and you say “mud volcanism” in the next paragraph to differentiate.

Deleted as suggested.

Lines 178-179: Add references re sector collapse at Mt Sidley and Mt St Helens.

References added.

Lines 180-182 (See also comments on Figure 5): I think more complete treatment of the NW flank of Jezero Mons would be helpful here. I see two large valleys or erosional troughs on the NW flank, from which dark, lobate deposits extend into Jezero crater. I recommend you add debris flow to your list of possible interpretations.

Line 181: Again, here I see an inconsistency relative to your response to Reviewer 2. If you want to call these flows (which seems reasonable), just make a case for it relative to a deposit with an erosional margin.

We removed the word “flows” (which was indeed disfavored by our original reviewer #2) and otherwise revised this sentence in line with this reviewer’s very helpful analysis and suggestions. It now reads “Two other erosional troughs on the mountain’s northwest flank appear to have sourced lobate deposits that extend into Jezero crater (Fig. 5b-c); these look smooth at meter scales, not unprecedented for martian lavas⁷⁹, but could instead be interpreted as lahar, debris flow or pyroclastic flow deposits.”

You should support your statement re the smoothness of the possible flows with references to the literature. There are many possible studies to choose from.

I disagree with the statement that these are unusually smooth for Martian lava flows; see this reference for treatment of rough (a’ a-like) and smooth (pahoehoe-like) lava flows in southern Tharsis:

Crown, D.A., and Ramsey, M.S., 2017, Morphologic and thermophysical characteristics of lava flows southwest of Arsia Mons, Mars, *J. Volc. Geotherm. Res.*, 342, Pattern to Process: Remotely Sensed Observations of Volcanic Deposits and Their Implications for Surface Processes, 13-28, doi:10.1016/j.jvolgeores.2016.07.008.

*We would argue that the potential flows into Jezero appear smoother at **HiRISE** scales than those near Arsia, even if both appear smooth at CTX scales... but the Jezero ones are not perfectly smooth either, and the general point is a good one. As noted just above, we revised the statement to “these look smooth at meter scales, not unprecedented for martian lavas,” with citation to Crown & Ramsey (2017).*

Methods

Lines 221-232: Re thermal inertia: It would be helpful to report the range and standard deviation for the thermal inertia values in Table 1 to provide better context.

Standard deviations have been added for all thermal inertia values in Table 1, and the text also now includes the full range of values measured across Jezero Mons.

Perhaps also include surface area in Table 1. This also raises the question of what boundaries you used for each of the volcanoes—can you provide a simple explanation or references to someone else’s mapped boundaries?

We have added the following sentence: “Boundaries for each mountain were delineated based on examination of visible and thermal images, resulting in measured surface areas of 584, 661, 574, and 470 km² for the Jezero, Zephyria, Apollinaris, and Thaumasia edifices, respectively.”

Lines 284-296: Re diameters and slope values: A bit more complete documentation would be helpful here. Most importantly, please include how each volcano’s boundary was determined. Re slopes, why did you use 15 different profile lines (and how were these selected) instead of averaging slopes from the N-S, E-W, NW-SE, and NE-SW lines used for other parameters?

We added here a note that boundaries were “delineated manually based on breaks in slope [citing one of the references recommended by this reviewer, which describes this as standard practice] and on its distinctive properties relative to surrounding areas in visible and thermal images,” and that the 15 profile lines were “equally angularly spaced, in order to produce a single representative number for direct comparison in spite of substantial azimuthal variations in some cases.” E.g., as you can see from looking at Jezero Mons in Figure 1, some sides have substantial valleys or ridges that could have unduly influenced the average flank slope value if we only averaged four profiles.

Perhaps, “Slopes” in Table 1 should be “Flank Slopes” given method?

Revised as suggested.

Table 2

I recommend you include Jezero Mons in the title of the table as I believe this table is site specific.

Revised as suggested.

Re Stratovolcano entry for low thermal inertia. One could reasonably expect the particle/block size characteristics of a stratovolcano to be highly variable (in both space and time) due to the combination of explosive and effusive eruptions that may have formed it. Thus, I would not conclude that low thermal inertia is a reliable indicator of a stratovolcano.

This is a fair point, and consistent with the other reviewer’s suggestion that we instead use the term “composite volcano,” reflecting a potential combination of explosive vs. effusive components as you note, with likely varying thermal inertias. In the scheme adopted for the table, we have therefore changed the check mark here to a question mark.

In general Table 2 seems overly simplistic/generalized. Size and shape could be expressed in more specific terms. Thermal inertia is problematic given uncertainty regarding erosional history

and potential deposition of mantling deposits. Re “Less cratered than surroundings”: The local geology is quite complex so it would help to be more specific here.

With respect, we feel that an admittedly simple presentation like this is the best way to help this journal’s broad audience visualize how the full suite of data considered herein leads us to tentatively favor the composite volcano hypothesis. By “size and shape” we largely mean the morphometry tabulated for Jezero Mons in Table 1—and so have now updated Table 2’s caption to say “Size and shape (morphometry)”—but it is different elements of size or shape that drive our assessment of consistency (or not) with each potential hypothesis: nearly all parameters in Table 1 appear more consistent with hypothesized martian composite volcanoes than with the shield volcano, whereas for mud volcanoes it is specifically the diameter of Jezero Mons that appears out of range, whereas for pedestal craters it is the flank slopes. All of this is discussed in more detail/nuance in the text, with the table then attempting to present the “bottom line.”

Re: thermal inertia, we think it is worth including despite the uncertainties because, as noted above, it is a parameter in which Jezero Mons stands out relative to most other circum-Jezero materials, which in itself seems an argument against the “null hypothesis” that the mountain is just one part of a widespread “massive basement unit.” The pedestal crater or shield volcano hypotheses are indeed only effectively refuted by thermal inertia if it can be trusted to measure “bedrock” properties (i.e., not unrelated mantling deposits), but we think it is reasonable to present the data and allow skeptical readers to ignore the “thermal inertia” column if they prefer (all our other disfavored hypotheses have at least one other “X” column anyway).

Re: “less cratered than surroundings,” we can’t readily think of a clearer way to express the point. Our original submission had this column labeled “low crater count,” which a reviewer said was too vague; but given the large uncertainties on our age estimate, we hesitate to quote anything more precise, such as “Amazonian age.” Additionally, a pedestal crater could in principle have any age, but as we note in the Discussion, “low crater density on Jezero Mons suggests that it is less resistant to erosion than its surroundings, rather than more resistant as expected for a pedestal crater.” So again, it is really the comparison to surroundings that matters; we based this on a qualitative comparison and did not do formal crater counts on any adjacent units that could be directly specified here, but our text does quote several other authors’ crater-based age estimates for Jezero’s fluvial landforms and crater floor, all of which are older than ours for Jezero Mons. But if the reviewer or editor can think of a specific alternate heading that would be preferable here, then we can change it again.

Figure 1:

Figure 1c caption: Do you need to change wording re “flows into Jezero crater” to be consistent with other revisions made re this in response to Reviewer 2?

Again, our prior omission of “possible” in this instance was inadvertent; we have now thusly qualified the “flows” wording.

I think Figure 1a has value but would also like to see a topographic profile as recommended by Reviewer 1 so I recommend that addition.

We have now added a topographic profile to the bottom panel of this figure.

Figure 2:

Figure 2a caption, Line 535: Change “arrows” to “arrow.”

We moved the comment about arrows to the start of the caption (before the individual panel descriptions) to clarify that it refers not only to the arrow in panel a but also the two in panel c.

Figure 2c: This image is too zoomed in to really appreciate the potential linear segments.

Yes, unfortunately the full summit crater is just a little bit wider than a single HiRISE image, so the full rim is not quite captured by the image as previously submitted. We have now updated it to include adjacent CTX image data so that the full crater is captured in 2c, and have also updated the text reference to this figure to point to both 2a and 2c for illustration of the linear segments, since the former shows a more zoomed-out view (which can be directly compared to the Mt. Sidley analog in 2b).

Figure 3 caption: Since the spectra are numbered, they should be referred to using the numbers in the figure caption (instead of “bottom two”).

Changed “bottom two” to “#1 and 2.”

Figure 4: The images are all HiRISE excerpts but have very different colors. Caption could be expanded to provide more complete explanation of these images. For example, Figure 4c looks like it includes both b/w and color HiRISE.

We have added the following sentence for further clarity (also updating reference 67 to one that describes these Merged IRB data products): “All panels show Merged IRB products⁶⁷ to display color where available (each stretched to enhance contrast) and otherwise RED data in grayscale.”

Figure 5: What I perceive to be the margins of Jezero Mons are quite interesting. I am not sure these three images are the best choices to show the nature of the margins of Jezero Mons and relationships with adjacent terrain. Consider showing additional HiRISE and/or CTX to more fully document the margins (e.g., the deposits within Jezero crater).

Please also state whether or not (or uncertain) you interpret the possible flow edge to be a primary lava flow front or an erosional margin in the text with appropriate supporting information. Do the margins of Jezero Mons have consistent morphologic characteristics around its perimeter?

The margins are not well exposed everywhere around the perimeter, but rather just “in several areas” as we note when first citing Figure 5 in the text. This and the still-limited availability of highest-resolution or color image coverage led us to choose the three images in the figure as the best we could find to illustrate the relationships with adjacent terrain, although we hope that additional coverage will allow further study in the future. Our highest-resolution view here (panel c) shows a jagged edge to the interpreted flow margin, at a scale comparable to the

spacing of adjacent aeolian ripples; we interpret this as evidence for at least some erosional modification of the flow margin and thus have added the word “eroded” to the caption here.

Figure 6: Suggested change for heading to Crater size-frequency distribution for flanks of Jezero Mons. Should specify in caption if the age you report is from fitting the whole distribution or some limited crater size range. It would also be useful to indicate in the caption at what crater size the CSFD appears to deviate from the isochron as this provides some indication of surface degradation.

Heading changed as suggested, and the caption now specifies that the age fit only applies to craters of ~400 m diameter or larger.

We again thank the editor and the two reviewers for their valuable further comments. Our responses are interspersed below, in italics; in all cases at least one related change has been made to the paper. Our only other change is to the departmental affiliation of one co-author, and those formatting changes required to complete the editorial requests checklist.

Reviewer #3 (Remarks to the Author):

No further comments on my part. I can recommend the article to the editor for possible publication.

We thank the reviewer for this positive assessment and for their helpful suggestions on our prior version.

Reviewer #4 (Remarks to the Author):

I have reviewed the revisions made to the manuscript and the authors' responses to the previous review comments. The authors have done a thorough and commendable job of responding and refining their manuscript accordingly. I offer a few comments below for the authors' consideration prior to publication.

We are grateful for all of the helpful comments on this and our prior version!

Lines 62-71 (original comment and line numbers): I appreciate the explanation offered and revisions made in response. I recommend one or both of the following re this point:

- a) Include in the Results section the substance of the review comment response in the manuscript to explain that the definition of the outer margin of Jezero Mons is not clear and why.
- b) Add a dashed line to Figure 1c to indicate the approximate boundary of the volcano with "?" inserted where the boundary is most uncertain.

We appreciate the reviewer's perspective on this! We have now added a dotted outline of Jezero Mons to the former Figure 1c (now 1e), as requested.

Lines 126-137 and Table 1: I think you should refer the reader to the Methods section re the morphometric parameters. The explanation offered is reasonable but I would still prefer to see even a rough outline of its base (as suggested above).

We added a citation to "details in Methods" at these lines in the text, and also added the outline to Figure 1e.

Table 1. Should there be citation noted for Uranius Tholus and Mt. Sidley on this table?

Citations to a relevant reference for each now added.

Table 2: I think you should probably define what the X's, ?'s, and boxes mean in a footnote.

We now added this information to the Table caption.

Figure 1:

a) Include color scale bar for Fig 1d.

Scale bar added.

b) The topographic profile is a nice addition. It would be useful to give an azimuth for the topographic profile or show the start and end points on Fig. 1e.

The start point would be hidden behind the mountain's summit in Fig. 1e, so we noted the azimuth in the caption: "transect starting ~68° east of north from summit crater's center."

Figure 5: Consider adding arrows to Figs 5a and 5b to show features of interest.

Arrows added to both.